

# Accessing Diverse Data Comprehensively - CODM the COSYNA Data Portal

Gisbert Breitbach[1], Hajo Krasemann[1], Daniel Behr[1], Steffen Beringer[1], Uwe Lange[2], Nhan Vo[3], and Friedhelm Schroeder[1]

[1]Helmholtz-Zentrum Geesthacht, Institute for Coastal Research, Max-Planck-Str.1, 21502 Geesthacht, Germany
[2]Brockmann Consult, Geesthacht, Germany
[3]smile consult, Hannover, Germany

*Correspondence to:* G. Breitbach
(Gisbert.Breitbach@hzg.de)

**Abstract.**

The coastal observation system COSYNA aims to describe the physical and biogeochemical state of a regional coastal system. The COSYNA data management is the link between observations, model results and data usage. The challenge for the COSYNA data management CODM[1] is the integration of diverse data sources in terms of parameters, dimensionality and observation methods to gain a comprehensive view of the observations. This is achieved by describing the data using metadata in a generic way and by making all gathered data available for different analyses and visualisations in an interrelated way, independent of data dimensionality. Different parameter names for the same observed property are mapped to the corresponding CF[2] standard name (Eaton et al., 2010) leading to standardised and comparable metadata. These metadata together with standardised web services are the base for the data portal.

## 1 Introduction

In the last years various portals for ocean integrated observing systems have been created like the Australian Ocean Data Network Portal[3] (Trull et al., 2010), the US Integrated Ocean Observing System[4] (IOOS, 2010), the Regional Ocean Observing Systems (ROOS) by Copernicus Marine Environment Monitoring Service[5] and EMODnet[6] or systems like PANGAEA[7] (Diepenbroek et al., 2002). The latter is a more general collection of finalised environmental data including the ocean data and is not focused on observing systems. Starting 2009 the Helmholtz-Centre Geesthacht build up COSYNA together with various partners[8].

---

[1]COsyna Data and Metadata, http://codm.hzg.de/codm
[2]Climate and Forecast
[3]http://portal.aodn.org.au/aodn
[4]http://www.ioos.noaa.gov/catalog/welcome.html
[5]http://marine.copernicus.eu
[6]http://www.emodnet.eu
[7]http://www.pangaea.de
[8]Alfred Wegener Institute, Bremerhaven; Bundesamt fuer Seeschiffahrt und Hydrographie, Hamburg; marum, Bremen; Institut fuer Chemie und Biologie des Meeres, Oldenburg; Niedersaechsischer Landesbetrieb fuer Wasserwirtschaft, Kuesten- und Naturschutz, Norderney; Hamburg Port Authority, Hamburg;





Most of the observing system portals follow the idea that data should be freely available to everybody. Data access and visualisation for observations is based on measurement platforms. That implies that the user first has to select a platform and can then access the data for that platform. The approach of the COSYNA data portal CODM is different. First one selects an
observed property (i. e. parameter) and the temporal-spatial area of interest and then comprehensive access to all data from different platforms is established.

Most portals of the big national observation systems like IOOS are linked to other portals of regional systems or portals of integrated systems for a single type of observation (e.g. high frequency radar[9]). CODM offers an integrated portal for all COSYNA observations and additionally for operational model results. Thus, the portal allows an integrated access to highly
diverse data with focus on the observed property.

Section 2 of this paper describes the goal and the objectives of CODM. The next parts present the general outline of data management in COSYNA to meet the objectives with a focus on the essential elements including various web-services. The implementation of these web-services together with metadata are a new feature of CODM allowing a flexible user adapted visualisation and retrieval of all searched data. In section 3.7 the integration of web-services and the concept of CODM is
described. Short sections on data quality from the viewpoint of data management and data policy follow. In section 6 some illustrative examples of the different ways of data visualisations implemented in the CODM portal are presented. Finally in section 7 very useful visualisation tools are described.

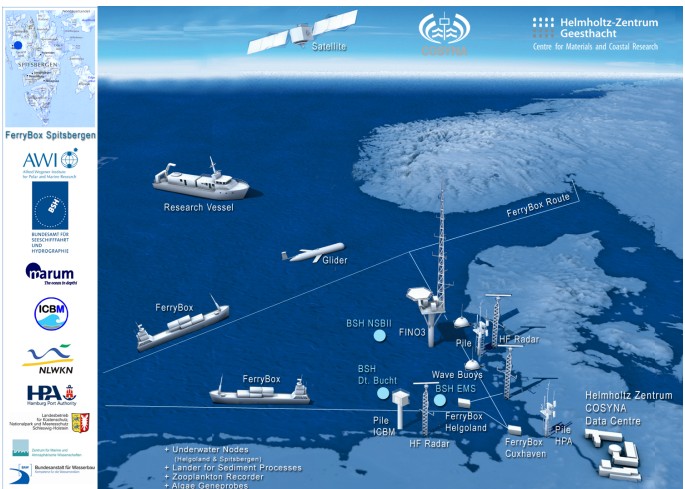

**Figure 1.** Main locations for COSYNA observations in the North Sea and at Spitsbergen (top left). The logos of COSYNA partner institutions are shown.

---

Landesbetrieb fuer Kuestenschutz, Nationalpark und Meeresschutz Schleswig-Holstein, Husum; Zentrum fuer Marine und Atmosphaerische Wissenschaften, Hamburg; Bundesanstalt fuer Wasserbau, Hamburg

[9]http://www.ioos.noaa.gov/hfradar/welcome.html





## 2   Goal of CODM

The objective of CODM is to gather often heterogeneous data from COSYNA in an integrative way. To achieve this objective all related data must be homogenised with regard to data structure and have to be combined in plots and maps for visualisation. One solution would be the application of an ontology like proposed by the Semantic Sensor Network Group (Lefort et al., 2011). For CODM the mapping to standardised observed property names, the CF standard names (Eaton et al., 2010), is a fast and less complicated solution. Within the metadata these standard names are mapped to the internal parameter names used by the scientists who setup the sensors and the data acquisition.

The user of the COSYNA data portal should be able to select an observed property and the spatio-temporal extent of interest based alone on information from metadata. There should be no access to data before a visualisation or download. Data access is performed solely via standardised web services.

The COSYNA data policy is free and open according to the idea that all data should be open to everybody without any restrictions or collection of personal data. The understanding of user requirements and the optimising of the system accordingly needs the monitoring of user access to COSYNA which is a conflicting objective to the open data approach.

## 3   CODM System Description

### 3.1   Observations

Observations in COSYNA result from different types of measurement devices leading to different types of data:

- At fixed positions

  - Buoys with CTDs (device measuring conductivity, temperature, pressure and more) and ADCPs (acoustic Doppler current profiler) at different fixed depths.

  - Waverider buoys

  - Stationary Ferryboxes

  - Under water nodes with CTDs and ADCPs.

- Moving platforms

  - Ferryboxes on ships

  - Gliders

  - Scanfish

- Remote sensing platforms

  - Satellites (Modis on Aqua, Meris on Envisat)



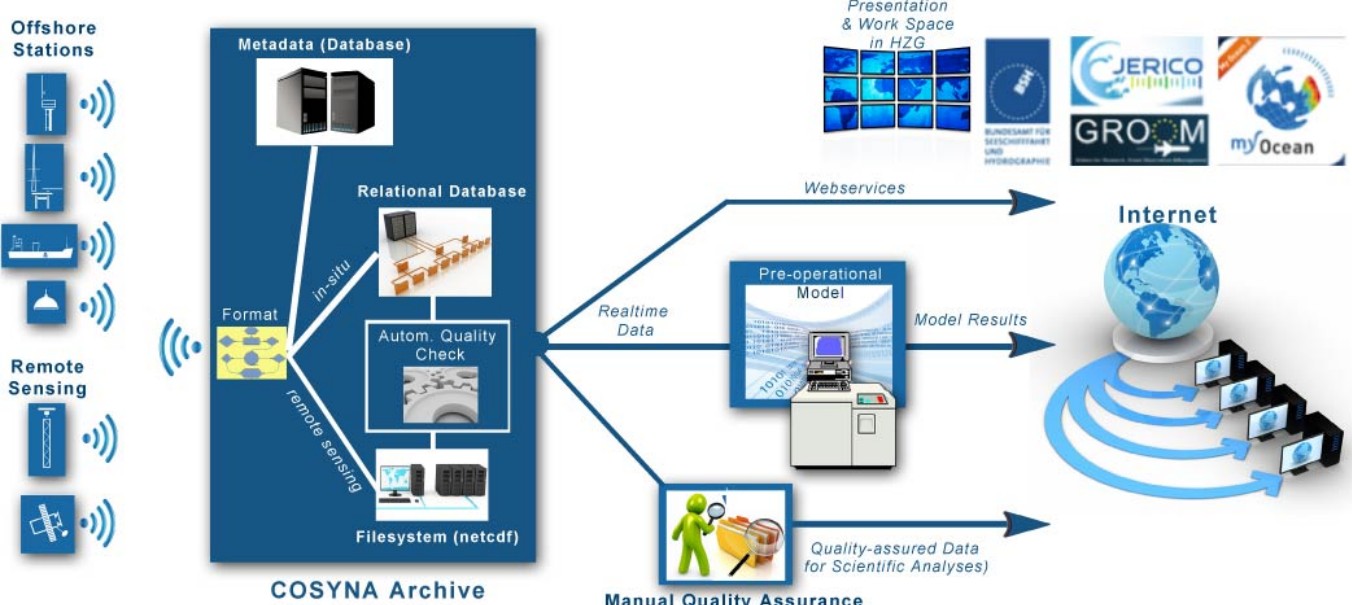

**Figure 2.** Data workflow in COSYNA

    – Landbased HF radar

    All observations will be described in other articles contained in the Ocean Science and BioGeoScience special issue entitled "COSYNA - Coastal Observing System for Northern and Arctic Seas" (Baschek et al., 2016). A schematic overview of these observations is shown in fig. 1.

### 3.2 Databases

As indicated in fig. 2 all in-situ observations are stored in relational databases (Oracle). For every type of in-situ observation a database with a data model suited to the observation type is used. Data from Ferryboxes operating on steady routes are stored in the database which can be accessed directly under ferrydata.hzg.de. Data from stations of fixed locations are accessible under tsdata.hzg.de. Survey data from ships not using steady routes can be accessed under surveydata.hzg.de.

### 3.3 Models

One goal of COSYNA is to integrate observations and numerical models to get a synoptic view of the state of the coastal areas. This integration is done by assimilating real-time observations into a model re-analysis. Another objective of CODM is to enable an online validation of these models using observation data which are not used for assimilation.



### 3.3.1 Circulation Model

Based on the GETM[10] model (Stips et al., 2004) data from HF radar observations are assimilated into a re-analysis of the currents in the North Sea (Stanev et al., 2011). In addition temperature data from OSTIA (Donlon et al., 2012) and Ferryboxes are assimilated into GETM.

### 3.3.2 Wave Model

Driven by data from DWD[11] a prognostic wave model is run which on reference provides wave parameters for every hour up to a 36-hour forecast (Behrens, 2009). In principle, the model output could be used for assimilation of observations too. In practice it is difficult to determine high quality wave parameters from HF-radar observations. On the other hand, the wave model data are so consistent with observation to a degree that renders it unnecessary to improve the model via data assimilation (compare fig. 5).

### 3.4 Collecting Observations

All COSYNA observations are collected in near-real-time. The principle of the data flow is shown in fig. 2. Data are stored as either netCDF files (Rew and Davis, 1990) in the COSYNA filesystem (remote sensing platforms) or as time-series in relational databases (all other sources). Metadata are also stored in a relational database. The netcdf-output of model calculations is treated just like the netcdf-files from remote sensing observations.

### 3.5 Metadata

The underlying concept for CODM was to build a data portal with a spatio-temporal search processed solely within the metadata. Real data are not accessed before visualisation or download occur. Hence, creation of metadata is crucial for the underlying concept. The automation of data handling, which ranges from data search to data display and data retrieval, relies on the stored metadata. It is necessary to use a harmonised vocabulary for the names of the observed properties and this is realised by using CF standard names (Eaton et al., 2010) which are mapped to the originally used notations. This is done internally within the metadata structure and ensures a common usage of parameters supplied by the primary data providing devices. The structure and content of the metadata are critical as they allow joined searches and retrievals of diverse data types. The COSYNA data portal which functions as a system of services is described in section 3.7. The COSYNA data portal application retrieves data and communicates internally by using metadata, not only for the measured data itself but also for the sensors used for collecting data. This necessitates that two types of metadata are used within CODM:

1. Metadata for describing devices, like observation platforms with sensors generating environmental data. This includes numerical models. Within CODM they are called *platform metadata*.

2. Metadata for describing observations in the coastal system but also including model runs. These are called *data metadata*.

---

[10]General Estuarine Transport Model
[11]Deutscher Wetterdienst - German Weather Service




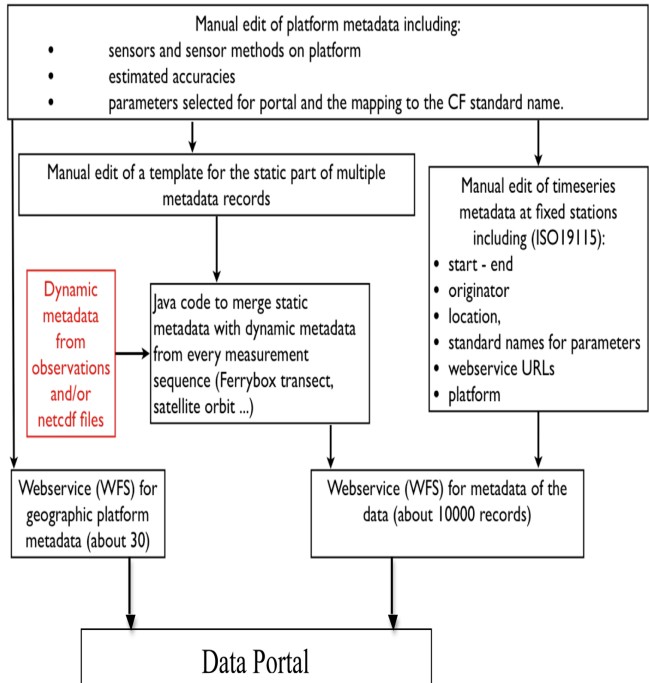

**Figure 3.** Scheme of the automatic metadata creation process.

**Platform metadata** contain the name of the platform including the data provider. Examples for platforms are buoys, Ferryboxes, wadden sea poles or HF-radar. A platform may consist of several sensors measuring one or many different observed properties. These sensors and the observed properties are described within the platform metadata. The geographic positions of platforms are stored in the metadata either as a point for stationary platforms or for platforms covering an area as a bounding box or encircling polygon. Also the start and end time of the respective platform operation is stored in the platform metadata. Furthermore, descriptions of measuring methods are stored. An example for a method to measure oxygen would be the oxygen optode. Platform metadata are complemented with ICES platform codes and WMO numbers, if existing.

For numerical models two groups of parameters are defined and described within platform metadata: Input parameters used to force the model on one hand as well as output parameters produced by the model.

The names of the internal parameters used in COSYNA are not harmonised but remain as they were chosen and used originally. A mapping of the original names to a standardised vocabulary is needed to ensure a common presentation and analysis of data from different platforms. The concept of parameter mapping is realised by introducing an additional, virtual sensor called *selectedparameters* as part of the platform metadata.





The additional sensor is not existing as a real sensor but carries CF standard parameter names which represent the measuring parameters. The observed property names of parameters measured belonging to this sensor are always CF standard names. The

internal sensor name and the name of the internal parameter are specified in the parameter description.

In the present context, this additional sensor just uses the metadata structure for a sensor to describe the mapping. With the help of this sensor a user can track the real sensor behind the corresponding CF standard name. This structure thereby allows for an interrelated search for comparable parameters in the portal as shown in fig. 1. A search thus combines various sources of measurements and model output to create an integrated view of data originating from different sources.

**Data metadata** include the start and end time of a measurement, the start and end location, graphic previews for the observed properties, if available, the person responsible for the data or the metadata and the URLs of web-services for visualising and downloading the corresponding data. In the case of platforms at fixed positions the data are described as time-series. This means that only one metadata record is needed for the whole time range covered by the platform. If the platform supplies data at multiple positions, a metadata record is created for every dataset. Datasets may originate from transect measurements, as

in the case of data from ships and gliders or a dataset may be represented by a single netCDF file. These multiple metadata records are created automatically following the procedure outlined in figure 3.

The COSYNA metadata are stored in the NOKIS (Lehfeld and Reimers, 2009) metadata system which is highly standardised. The data metadata are compliant with ISO19115 (ISO19115, 2003) as well as with INSPIRE[12] (INSPIRE, 2007). The structure for platform metadata have been developed within NOKIS. A migration to SensorML metadata (Botts, 2007) is being

considered.

## 3.6   Web-Services

Web-services are used to visualise or to download data. The details of their usage are kept within the metadata for each measurement allowing the CODM portal not only to link to a web-service but to execute the user-request and deliver the data or plot as described in more detail below. For data stored as netCDF files they can be downloaded via OPeNDAP[13] (Cornillon

et al., 2009). If netCDF files correlate with area data they can be visualised as OGC-WMS[14] maps (WMS, 2004) with the help of ncWMS (Blower et al., 2013). In addition, the versatile tool ncWMS is able to create time-series plots at selected positions within the represented area of the netCDF file.

The presentation of data from moving platforms such as FerryBoxes, gliders or ships needs additional effort. A WMS servlet was coded in Java and added as web-service which is producing and presenting colour coded transect maps of the measurements made by moving devices. Parameter plots of time-series at fixed platforms can be visualised by web-services using an application with direct connection to the COSYNA time-series database TSdata. A similar application is used to build parameter plots for transects. Downloads for all data stored in the Oracle database are provided through the software PySOS[15]

---

[12]Infrastructure for Spatial Information in the European Community

[13]Open source Project for a Network Data Access Protocol

[14]OpenGIS®Web Map Service Interface: Standard provides a simple HTTP interface for requesting geo-registered map images http://www.opengeospatial.org/standards/wms

[15]http://sourceforge.net/projects/pysos/



**Figure 4.** Concept for the interaction between users of the portal, data and metadata. All interaction is done using various web services. The data itself are stored as netCDF files or as rows in the Relational Database Management System (RDBMS) Oracle



an implementation of a Sensor Observation Service (OGC-SOS) (Na and Pries, 2007) which has been adapted for Oracle. The

standard OGC-SOS is part of the Sensor Web Enablement framework (OGC-SWE) (Botts and Reed, 2006) which improves the interoperability between sensors. CODM uses OGC-SOS not at the sensor tier currently however one abstraction level higher at the database tier.

The Web Processing Service (OGC-WPS) (Schut, 2007) PyWPS[16] is used to create additional services. For example a service which transforms SOS xml-output to a human readable ASCII table is provided as well as a service to plot wave energy

against time and frequency for wave rider buoys. Except of plots and WPS services, all services are provided by Tomcat web servers (Brittain and Darwin, 2008).

A Web Feature Service (OGC-WFS) (Vretanos, 2002) is provided using the open source software Geoserver (Geoserver Project, 2001) to provide access to metadata of platforms and real data as 'web features'.

### 3.7 Integration of Web Services

The COSYNA metadata system is based on NOKIS (Lehfeld and Reimers, 2009). The metadata can be accessed using the Catalog Service for the Web (OGC-CSW) (Nebert, 2007). In addition, a catalog service based on a Web Feature Service (OGC-WFS) was created which is optimised to be used by a data portal. As this metadata service supplies the URLs to access the corresponding data as downloads, maps or other visualisations, the COSYNA data portal CODM is capable of offering all types of available web-services to the users. The diagram in figure 4 gives an overview of the portal, its substructure, data

access and possible user interactions.

A unique feature of the integration of web-services in CODM is the storage of web-service URLs in the metadata allowing the direct use of them. Some other approaches like NOOS[17] are storing web-service URLs as well but only as general GetCapabilities-URLs. These general URLs provide information about the web service but cannot be used to access the data directly. CODM stores URLs which leads to immediate data access as map, plot or numerical download. These web-service

URLs have a static part and a dynamic part. The dynamic URL parameters are defined using a xml description. An example for a Web Map Service is shown in listing **??**.

In general, any data portal may use these mechanism of building web-service URLs. This is depicted in fig. 4. The user interface could be CODM or another portal. Any portal can access COSYNA data because all the web-services are freely accessible. EMODnet physics (Novellino et al., 2014) uses the web-services of HF-radar stored in CODM to create the web-

service for visualisation of the currents in the German Bight[18].

CODM provides various search options to its users. The entry point for each refined search is the selection of the desired observed property, a time and depth region and a geographic area of interest (fig. 1). After clicking "Select all datasets" one of the buttons "Create map", "Create plots" or "Downloads" can be used. This means the requested data are provided with just 3 clicks. As an example figure 2 shows the output of a click on the "Create map" button for Chlorophyll measurements derived from MERIS and FerryBox data for the end of April 2009.

---

[16]http://pywps.wald.intevation.org/documentation/
[17]http://www.noos.cc
[18]http://www.emodnet-physics.eu/map/FeedPlatformInfo.aspx?id=12179



```
<mapService>
 <name>ncWMS</name>
 <baseurl>
 http://codm.hzg.de/ncWMS/wms?REQUEST=GetMap&VERSION=1.3
 .0&STYLES=&CRS=CRS:84&WIDTH=256&HEIGHT=256&FORMAT=image
 /png;mode=32bit&TRANSPARENT=true&LAYERS=11/eastward_vel
 ocity&BBOX=&elevation=&time=&COLORSCALERANGE=
 </baseurl>
 <dynamicParameters>
  <time>
   <name>time</name>
   <default>2011-12-11T00:04:27.000Z</default>
   <syntax>yyyy-MM-ddThh:mm:ss.milZ</syntax>
  </time>
  <valueRange>
   <name>COLORSCALERANGE</name>
   <default>-1,1</default>
   <min /> <max />
   <syntax>min,max</syntax>
  </valueRange>

   <name>BBOX</name>
   <default>
    5.91667,53.418142548596116,8.970867584195927,55.2
   </default>
   <latmin>-180</latmin> <latmax>180</latmax>
   <lonmin>-90</lonmin> <lonmax>90</lonmax>
   <syntax>latmin,lonmin,latmax,lonmax</syntax>

  <depth>
   <name>elevation</name>
   <default>0</default>
   <syntax>value</syntax>
  </depth>
  <log>
   <name>logscale</name>
   <default>false</default>
   <possibleValues>
    <logarithmic>true</logarithmic>
    <linear>false</linear>
   </possibleValues>
  </log>
 </dynamicParameters>
</mapService>
```

**Listing 1.** Example XML-listing for a WMS request

5    CODM can be accessed directly with the URL http://codm.hzg.de/codm.

In addition, the user can view the platform metadata as well as the metadata of a measurement by clicking on the icon in front of each list entry.

## 4 Data level and quality control

COSYNA data level definitions are based on the data level definition used for remote sensing data (Parkinson and King, 2006) but are expanded to include in-situ data. They start from level 0 for raw data to level 4 for externally published data. The definitions for the different levels are indicated in table 1. Data of level 3 and higher are available via CODM.

5    Level 3 denotes that the data have a defined unit and are geo-referenced. A quality control flag is applied to the data of level 3. A subset of the SeaDataNet (SeaDataNet, 2010) quality flags is used as a flagging scheme as shown in table 2. Externally published data have a final delayed mode quality control and for example are published in PANGAEA (Diepenbroek et al., 2002).



**Figure 1.** View of the CODM portal with the selected parameter chlorophyll-a and a selected time range from April 2009 to May 2009. After clicking "Select all datasets" the count of datasets for all platforms are shown and automatically selected. OpenStreetMap (OSM) (Coast, 2004) is used as background map. To keep the portal simple for the users well known parameter names are used instead of CF standard names



| Data level | Characterised by | Data-formats | Metadata | Processing steps to reach level | Data-quality status |
|---|---|---|---|---|---|
| level-0 (raw) | raw data from instrument, instrument units, only for experts. These data are often not accessible for standard users of an instrument and the scientist may have to rely on the instrument to supply ready level-1 data | raw-data (counts, bits, bytes, binary,...) | instrument-identifier, time (absolut or relative) | none or proprietary software | only external sources available (logbook,...) |
| level-1 (physical values) | bio-, chemical-, physical-values, data related to instrument, values in standard units, for use of specific scientific group related to instrument/device | ASCII, or data-specific physical, chemical, or biological parameters | time as it is delivered from the instrument. In case this is configurable follow level-2, position information variables (names, units) | instrument calibration, synchronisation between instruments (depth, gps) | corrupted data eliminated (i.e. data with missing bits, only partial numbers or tuples) |
| level-2 (bio-, chemical-, geo-physical value in space and time with error) | bio-, geo- physical-chemical values, connected to space and time of measurement, for use in scientific community of specific subject, correctable by version whenever faults were eliminated. | ASCII, netCDF or data-specific (physical parameters for objects of interest, e.g.: reflectance calculated from radiance, irradiance measurements) | time - preferable in (fractional) days since 1970. Position in preferable lat, lon (WGS84) and depth. Errors/measurement uncertainties | producing geo-physical quantities for object of interest by combining several level 1 variables (by binning, averaging over short distances or short times or other procedures) | quality values and/or quality flags connected to data-values are derived. Spikes, bad data flagged |
| level-3 (data-exchange: value in space-time with error in standard format) | values located in standard formats (e.g. common described grids) with error/confidence level, for use in general scientific community - in general portals accessible, correctable by version | preferable in netCDF for maplike, RDBMS for time-series like data | metadata in standard system. | map projected on regular oriented grid. Time in (fractional) days since 1970 (preferred). If applicable on time interpolated grid | the intention is to attach errors for every measured value. The data quality flag is filled. Link to descriptive meta-data in standard system |
| level-4 (published) | data published, kept in archives, preferable with DOIs, fixed dataset, cross-linked to eventual publication, correction of data results in new dataset, use by full scientific community | as needed by receiv-ing/publishing data centre (WDC) | as level-3 plus mandatory metadata on data quality | extended analysis performed to understand accuracy and limitations of the data-set in extended space and time dimensions (e.g. check for longterm trends and errors) to the largest extent possible | data quality controlled, all checks passed. Reviewed data set, sent to an WDC data centre with a DOI |

**Table 1.** Data level definition used in COSYNA. Some aspects like accuracies are presently not realised.



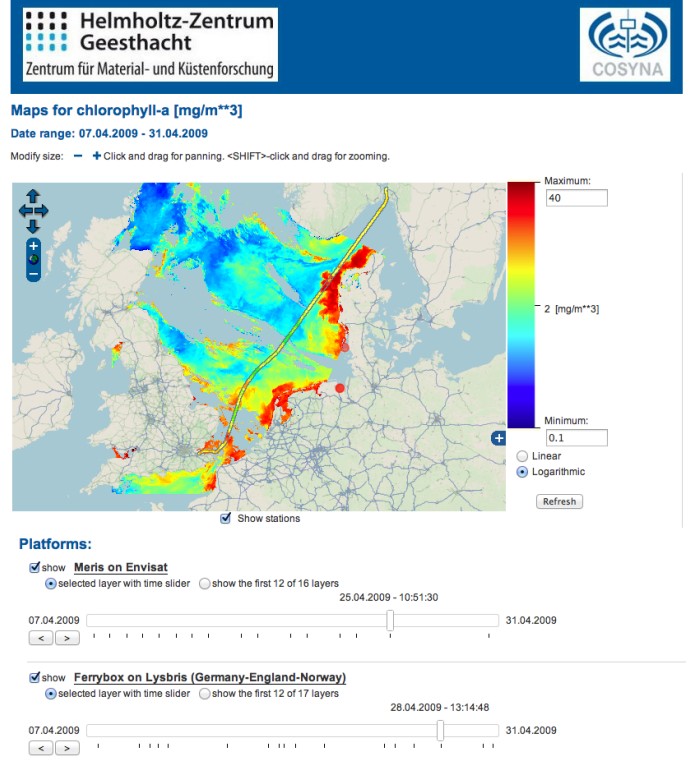

**Figure 2.** Comparison of a chlorophyll map derived from MERIS with FerryBox measurements both for end of April 2009. The MERIS data were deduced following refernce (Doerffer and Schiller, 2007). The concentrations are computed on a logarithmic scale. Ferrybox data are taken from reference (Petersen, 2012)

## 5 Data policy and access to CODM

COSYNA supports an Open Data policy. It is possible to download COSYNA data via CODM. Some guidelines for a fair data use are written in the COSYNA data disclaimer[19] which pops up before a download is started.

In addition COSYNA starts to publish data via the peer reviewed data journal Earth System Science Data (Carlson and Pfeiffenberger, 2009). This external data publication applies to COSYNA data level 4 with final quality control.

COSYNA data policy stipulates that data access should be unhindered. On the other hand, there has been increasingly interest from COSYNA funding sources to gather information on who is accessing and using COSYNA data. This conflict is solved by an open user registration process which defines user accounts completely based on user input without requesting personal information. Only self defined user name and password, country, city and user-category is asked as mandatory. The user-category is selected from a pre-defined list[20]. The self defined username password combination is needed to access CODM.

---

[19]http://www.coastlab.org/Disclaimer.html

[20]Science, Private Businesses, Industry, Public Administration and Authorities, Politics, Interest groups, General Public, Media and Education





| Quality Flag Value | Definition | Description |
|---|---|---|
| 0 | no quality control applied | so far no quality check tests have been performed on this data point |
| 1 | good quality | all available quality tests indicate that this point is of good quality |
| 2 | probably good quality | part of the available quality tests indicate that this point has good quality but some minor tests have been not or could not be applied so far. Best possible quality flag for real-time data points for which only automatic checks could be applied. |
| 3 | probably bad quality | some quality checks failed put this data point has the potential for its quality to be increased |
| 4 | bad quality | quality checks failed. This data point is not reconstructable |
| 9 | missing data | the data value is missing, quality checks couldn't be applied |

**Table 2.** Quality flag values used in COSYNA

The user registration process creates a connection between the user information and the IP-number of this user. With every new login this connection is renewed. Based on this connection all log files of web-service requests and responses can be analysed to gather the information about the usage of CODM. This analysis started in November 2014. The data downloaded per category for the accumulated first year and 2 month are shown in fig. 3.

The largest share of data access requests originate from the science category followed by administration. Only minor access requests stem from the general public. This result is not surprising because the available data and visualisations are targeting mostly science and administration. To address the general public a portal is needed with less data variety presented but more explanations and user guidance.





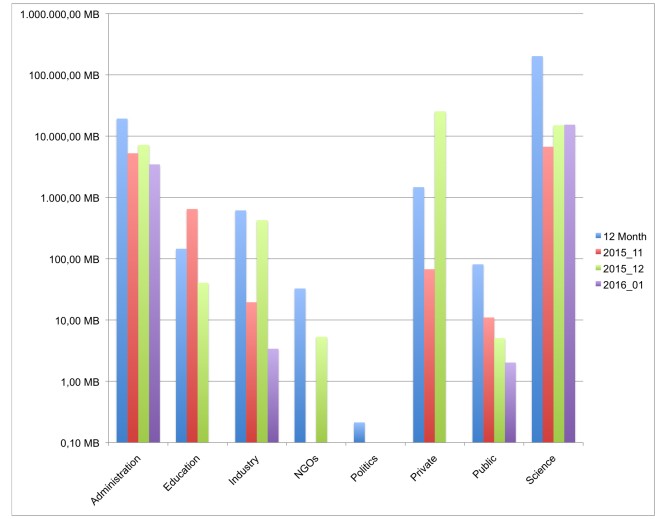

**Figure 3.** COSYNA data downloads since November 2014 per category. The category names are self explaining with the exception of "Private" which stands for private businesses like fishermen etc. 12 Month (blue) stands for the period November 2014 until October 2015.

## 6 Examples

To compare output from the forecast wave model WAM (Behrens, 2009) with observations, data for wave height during a winter storm are shown. The selected time range starts on 2013-12-01 and ends on 2013-12-08. A map of the wave heights from model results during the storm is presented in figure 4. In this map the positions of wave rider buoys are marked as red dots.

   The comparison is done with a click on one of the buoys marked on the map. One result for the wave rider buoy near
Helgoland is shown in fig. 5.

   Similarly good matches could be reached at the other measurement stations. Because the tools for creating the graphs of the forecast model (ncWMS) and the time series differ the plot layouts differ slightly.

## 7 Additional visualisation tools

Although all the objectives mentioned in section 2 could be met by CODM additional visualisation tools are required to get a more comprehensive view for some data. For example comparing measured HF radar data (Seemann et al., 2011) for surface water currents with those of model output for the same parameter using the portal is not a trivial exercise in that two or more
datasets covering the same area cannot be displayed simultaneously on a single map. To allow such a comparison a separate web application for data of equal extent was developed. This application uses synchronised maps to visualise datasets for a selected timestep. Figure 6 shows the measured HF radar current vector (right map), the GETM model run (left map) and the reanalysed model with assimilated data (center map). Another feature of this tool is the creation of time-series plots at a clicked



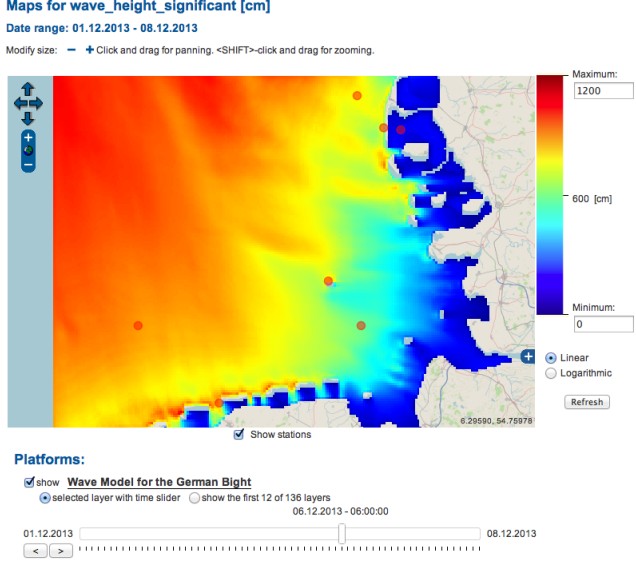

**Figure 4.** The map of the significant wave heights during the winter storm from a forecast wave model. (Data from (Behrens, 2014))

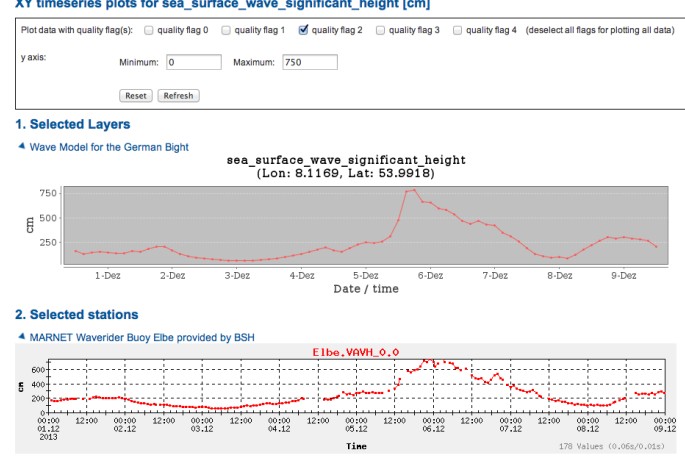

**Figure 5.** Comparison of the forecast model results (top) and the measurement of the significant wave height at the Helgoland wave-rider buoy during December 2013 (bottom). (Data from reference (Herklotz, 2014) and reference (Behrens, 2014))





**Figure 6.** Application comparing results for current vectors from the GETM model run without assimilation (left, data from reference (Staneva, 2015)), HF radar data (right, data from reference (Horstmann, 2015)) and HF radar data assimilated into GETM model results (middle, data from reference (Schulz-Stellenfleth, 2015)). Time series plots of the selected day are shown added below each map. The cross marks the position of the time-series.



map location. Shown in the lower part of figure 6 are the current direction components for the corresponding dataset on the

selected date.

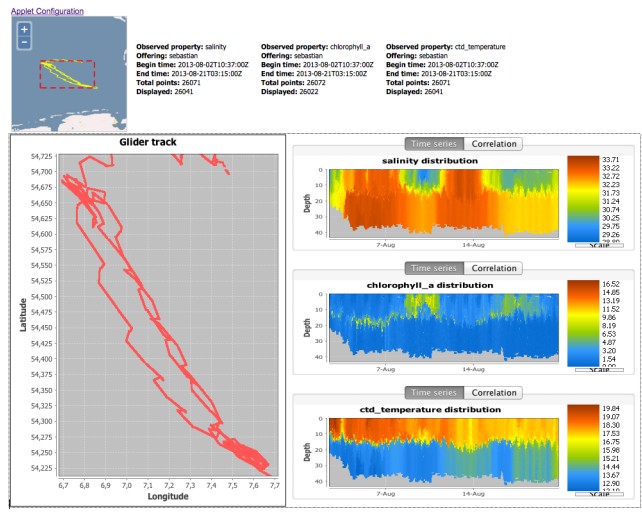

**Figure 7.** Java applet to visualise glider data in 3-dimensions. Three Parameters are selected. It is possible to zoom into location or into time. With "Applet Configuration" other parameters can be selected.

One basic presentation form of the COSYNA data portal are two-dimensional maps. However, a data search can also filter for depth ranges which are then applied to the data request using the metadata. For a few data types, such as glider data, it is useful to have a tool to visualise the data in three dimensions. A Java applet was built to display the location and the depth dependency of up to three parameters from a glider campaign (fig. 7). The URL for the selected observed property is stored

within the metadata similar to that of listing **??**. With this mechanism the tool is accessible as an additional icon from CODM. The further applet processing is independent of CODM. Up to three observed properties can be visualised with the applet. Beside glider data the applet can be applied to scanfish data as well.

## 8   Conclusions and Outlook

Modern marine observing systems are composed of many different observation platforms. This paper describes an approach for

the challenging task of integrating these various observations into one common portal while providing the ability to visualise the data in a concerted way. the CODM data portal demonstrates that it is possible to integrate in-homogenous observations and model output comprehensively. Furthermore, an online comparison of a data independent model, observed data and a data-assimilated model is provided. In addition, a solution for the challenging task of visualising data tracks in 3 dimensions has been developed.



This approach has been already realised in the COSYNA data portal CODM[21].

CODM is based on web-services and metadata. Unique features of this approach are the storing of web-service URLs as metadata as well as mapping observed or modelled parameters to standardised names. This enables the portal to present an integrated view and to compare different data sets and methods without any additional effort.

In the future new platforms with sensors accessed by a system using Sensor Web Enablement (Botts and Reed, 2006) to

access the observations will be available. Metadata for these new sensors will be described based on SensorML.

To gain information about usage and user interactions a registration procedure has been implemented in the portal. Only registered users are able to browse, view and download data. To comply with COSYNA's Open Data policy, registration is unrestricted, free of charge and without verification of the information provided by users.

The COSYNA data portal as well as many COSYNA web-services are registered in GEOSS[22] (Lulla et al., 2014). GEOSS

promotes common technical standards enabling data from thousands of different instruments to be combined into coherent data sets. The approach described here makes it easier for any system like GEOSS to integrate many data sources, thus ultimately creating a real earth observation system.

A deficit common to all existing approaches is the dependancy on observation platforms and different data types. It should be possible to integrate all observational data into a common data cube with a time dimension coordinate, three spatial dimension coordinates and one more dimension coordinate for the observed property. Such a data cube with the ability of homogenous

storing of various observational data should provide arbitrary cuts in all dimensions in a performant manner. As a start to realise the vision it is planned to integrate all COSYNA observations into a data cube in the near future. When the results are promising for COSYNA data there should be no obstacle to consecutively integrating more diverse data.

*Acknowledgements.*  The authors would like to thank Holger Brix and Frank Sellerhoff for the fruitful discussions we had about this work. We also acknowledge Desmond Murphy for providing valuable comments and assisting with language editing. Various COSYNA partners

inside and outside HZG delivered data which are used as examples in this paper.

---

[21]http://codm.hzg.de/codm
[22]Global Earth Observation System of Systems





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
