# Peer review of "Accessing Diverse Data Comprehensively - CODM the COSYNA Data Portal"

_Ocean Science, 2016_

## Referee Comment (RC1) · Anonymous Referee #1 · 9 Mar 2016

I am sure that the authors have done a great job in setting up the COSYNA data portal and getting the system running. What is missing is the localisation of CODM in the national and international landscape. I would like to ask the authors to give a better description on that.

Please also note the supplement to this comment:
http://www.ocean-sci-discuss.net/os-2016-6/os-2016-6-RC1-supplement.pdf

———————————————

[Figure]

**Supplement:**

[revised manuscript text omitted]

---

## Referee Comment (RC2) · Anonymous Referee #2 · 18 Apr 2016

This paper addresses a highly relevant topic: The interoperable access to coastal observation data. Especially the common approach to access very diverse data sets is very interesting. There are only a few aspects which should be addressed:

Page 2, lines 1 - 6: Here you describe that COSYNA offers a different way how to select data. It would be great if you could explain this decision a bit more. The approach to first select the observed property and then the spatio-temporal extend is useful. However, there may also be use cases in which the selected based on the platform as the first criterion might be useful. Would it make sense to support both approaches? The underlying SWE standards could allow to support both.

Page 7, line 10: Are the data metadata really related to single measurements (as you write) or to a time series as a whole?

[Figure]

Page 18, line 15: The reference to the listing seems to be broken.

All SWE standards that you are mentioning are used in their 1.0 versions. However, for some years, the SWE 2.0 specification are already available. It would be great if you could explain your plans if you want to upgrade to the SWE 2.0 standards which would offer some advantages (hierarchical structure of SensorML descriptions, more efficient SOS metadata, etc.).

You are explaining the COSYNA Open Data policy. It would be great if you could explain if you are using a specific license for publishing the data. Using such licenses gives users a higher level of security which constraints and conditions need to be considered.

The use of a WFS server for discovery functionality is interesting. It would be great if you could provide more explanations for the decision to use the WFS instead of the OGC Catalogue which is usually the typical interface for discovering resources.

Please have a general review of the spelling and grammar. For example the style of writing "NetCDF" should be harmonised (netcdf vs. netCDF). Also OGC standards should be written without a "-" (e.g. OGC WMS instead of OGC-WMS).

––––––––––––––––––––––––––––

---

## Author Comment (AC1) · 9 May 2016

Reply to: I am sure that the authors have done a great job in setting up the COSYNA data portal and getting the system running. What is missing is the localisation of CODM in the national and international landscape. I would like to ask the authors to give a better description on that. Please also note the supplement to this comment: http://www.ocean-sci-discuss.net/os-2016-6/os-2016-6-RC1-supplement.pdf

You asked for a localisation of CODM in the national and international landscape. As far as we know there is no review article about data portals so far. We think it is a good idea to write such a review about existing data portals. Only with such a review the national and international landscape of data portals can be defined. The authors of such an article should be either neutral or should represent a lot of different portals

from all over the earth. In our article we tried to locate CODM as far as we can do. 5 years ago we found that no existing portal was able to fulfil our requirements so started our new development. As far as we can see no other portal is still able to do what CODM is doing especially the inclusion of GetMap or GetObservation commands into the metadata. Another aspect that we don't know of any detailed description of other portals like it is done for CODM in this paper.

Comments to your 'in text comments':

Abstract line 8(Isn't that the aim of all data portals? Also, web services are standard for most other ocean data portals.): As worked out later in text the aim of CODM is not just making all gathered data available like most other portals. CODM is able to visualise data on an interrelated way by using web services. Most other portals uses the web service GetCapability-request CODM uses the GetMap- or the GetObservation-request.

Line 14 (Please, mention IMOS as well if you talk about Australian Data Centers.): This will be done.

Line 15 (PANGAEA is involved in a number of EC funded projects that focus on observing systems like FIXO3.): Nevertheless PANGAEA is not focused on observing systems with near-real-time data. PANGAEA is more interested on finalised data.

Page 2 line 10 (It is not just CODM that follows that approach.): This is not the assertion.

Line 18 (I am actually missing here a more throughout description of the ocean data portal landscape and the strategy to integrate them. A number of EC funded projects like seadatanet or ODIP shall be considered here.): This would be the task for a review paper about data portals. For this paper describing CODM in detail such a review is beyond the scope.

Page 3 line 8 (And at the same time a more rigid implementation that will probably lead

to extra efforts dealing with the diversity of data): There might be a disadvantage but for COSYNA data we can say that no extra effort is needed.

Line 14 (How do you resolve this conflict?): This is described later in the paper.

Line 17 (What is the intention of this section? It is neither a complete list of COSYNA observations nor is it clear how this influences the architecture of CODM.): It is a complete list of COSYNA observations included into CODM. All automatic COSYNA observation are included.

Page 6 line 10 (Isn't it a trivial statement? This entire paragraph needs to be revised.): We will revise this paragraph.

Page 7 line 17 (What does "highly standardised" mean?): In this case it means IN-SPIRE and ISO19115 compliant. The text will be adapted.

Line 20 (A few more words about the migration to SensorML would be vaulable here.): We are waiting for the results of ODIP2 before we could really decide to migrate to SensorML. In this paper we are not able to describe a migration which is not yet decided to do.

Page 9 line 30 (Is there a link between EMODnet and CODM?): The described connection is more than a link. It is an integration of COSYNA data into EMODnet.

Page 10 listing (Is this listing really needed?): We think that it is really hard to understand the emersed difference of CODM without this listing. Here it is shown that a mapservice is included as GetMap-request and how a portal could access the services in a syntactical correct way. We added more description in the figure caption.

Line 7 (This belongs into the user manual not in a publication): The passage will be removed.

Page 12 table (Is this an standardised scheme or just COSYNA specific? References?): As indicated in text this scheme is based on "data level definition used for

remote sensing data (Parkinson and King, 2006"). The definition were expanded to include in-situ data. The expansion is COSYNA specific but is adopted by MaNIDA for example. We reiterate the reference in the figure caption.

Page 14 table (Is there a reason to abstain from using the IOC quality flagging scheme? It is almost the same but at least a reference should be given here.): The IOC flagging scheme and the SeaDataNet scheme are different. The quality scheme is taken from SeaDataNet with slightly different definitions. This is indicated in text but will be included in the figure caption too.

Page 15 line 9 (Is this section really needed?): We think that examples are really needed for a data portal to make clear the advantages. Especially in a journal like Ocean Science where most readers are interested in the possibilities and accessible data of CODM and not so much in the technical methods. A data portal is not only a technical solution it is mainly a way to access data. Therefore examples how to access the data are needed from our point of view.

Some (Reiteration from above): We will try to avoid some reiterations but we think that some are useful.

---

## Author Comment (AC2) · 11 May 2016

Page 2, lines 1 - 6: Here you describe that COSYNA offers a different way how to select data. It would be great if you could explain this decision a bit more. The approach to first select the observed property and then the spatio-temporal extend is useful. However, there may also be use cases in which the selected based on the platform as the first criterion might be useful. Would it make sense to support both approaches? The underlying SWE standards could allow to support both.

The decision is based on the integrative aspect of COSYNA. There are specialised data portals to access time-series data, survey data, data of FerryBoxes on ships going on fixed routes and remote sensing data. These data portals should be used preferable if the platform is the first criterion. The main scope of CODM is the integrative aspect of

presenting data from different platforms together.

Page 7, line 10: Are the data metadata really related to single measurements (as you write) or to a time series as a whole?

Data metadata describing a stationary time-series are related to this time-series as a whole. Start time is the first value of the time-series. The end time lies in future. All other data metadata describe the whole measurement series e.g. a transect.

Page 18, line 15: The reference to the listing seems to be broken.

This will be corrected.

All SWE standards that you are mentioning are used in their 1.0 versions. However, for some years, the SWE 2.0 specification are already available. It would be great if you could explain your plans if you want to upgrade to the SWE 2.0 standards which would offer some advantages (hierarchical structure of SensorML descriptions, more efficient SOS metadata, etc.).

To be honest the SOS version is the pre-version 0.7 because the development of pySOS was stopped then. The conditions to be able to migrate to SensorML are described in the revised paper now. We are working on a SOS V2 solution as well. But it is not clear that such a solution can be integrated into CODM as it is described in the paper.

You are explaining the COSYNA Open Data policy. It would be great if you could explain if you are using a specific license for publishing the data. Using such licenses gives users a higher level of security which constraints and conditions need to be considered.

We are not using any licenses up to now but we are considering to do it in the future.

The use of a WFS server for discovery functionality is interesting. It would be great if you could provide more explanations for the decision to use the WFS instead of the OGC Catalogue which is usually the typical interface for discovering resources.

[Figure]

Main reason are performance and usability. A WFS could be configured just for discovery. A catalogue should contain all necessary information.

Please have a general review of the spelling and grammar. For example the style of writing "NetCDF" should be harmonised (netcdf vs. netCDF). Also OGC standards should be written without a "-" (e.g. OGC WMS instead of OGC-WMS).

This will be corrected.

---

## Author Response (AR2)

Response to report 2 from Ref. 2:
Instead of "web-service" the correct version would be "Web service" with a capital "W".

Is corrected

As suggested in the review you have harmonised the way how to write NetCDF. However, please check if "netcdf" is correct. It seems that "NetCDF" is the correct version.

Changed to netCDF.

Please check the usage of the word "like". In some cases this should be replaced by "such as".

Correctes at some places.

Page 3: Could you please add a short sentence explaining the term "stationary Ferryboxes". As Ferryboxes are described as sensor systems attached to ferries (which are moving), the meaning of stationary Ferryboxes should be explained.

A footnote is added.

Page 6: Before the citation of ISO19115 there is a missing space.

This is corrected

Page 6: The reference Botts, 2006 is outdated. It should be replaced by a reference to SensorML 2.0

New reference to Botts and Robin (2014) is used.

Page 7: The words "these mechanism" should be changed to either "this mechanism" or to "these mechanisms"

"These mechanisms" is used now

Response to Topic Editor:
Comments to the Author:
Please consider the recent response of reviewer #2.
It would have been also helpful to consider the general comment from
reviewer #1 to refer to more international projects...not necessarily in
citing a review paper but in citing individual publications or project websites...
but I finally leave this up to you. I consider this manuscript as a valuable add-on to this special issue and therefore recommend to publish it in this SI.

Added are a few sentences with two more references.